# Involuntary Full- and Part-Time Work: Employees' Mental Health and the Role of Family- and Work-Related Resources

**Deborah De Moortel** [1,2,*] **, Nico Dragano** [3] **and Morten Wahrendorf** [3]

1   Research Foundation Flanders, Egmontstraat 5, 1000 Brussel, Belgium
2   Interface Demography, Departement of Sociology, Vrije Universiteit Brussel, Pleinlaan 2, 1050 Elsene, Belgium
3   Institute of Medical Sociology, Medical Faculty, Heinrich-Heine-University Düsseldorf, P.B.101007, 40001 Duesseldorf, Germany; Dragano@med.uni-duesseldorf.de (N.D.); wahrendorf@uni-duesseldorf.de (M.W.)
*   Correspondence: deborah.de.moortel@vub.be

**Abstract:** Resources related to a good work-life balance may play an important role for the mental health of workers with involuntary working hours. This study investigates whether involuntary part-time (i.e., working part-time, but preferring full-time work) and involuntary full-time work (i.e., working full-time, but preferring part-time work) are associated with a deterioration of mental health and whether family- and work-related resources buffer this association. Data were obtained from the German Socio-Economic Panel (GSOEP) with baseline information on involuntary working hours and resources. This information was linked to changes in mental health two years later. We found impaired mental health for involuntary full-time male workers and increased mental health for regular part-time female workers. The mental health of involuntary full-time male workers is more vulnerable, compared to regular full-time workers, when having high non-standard work hours and when being a partner (with or without children). Involuntary part-time work is detrimental to men's mental health when doing a high amount of household work. This study is one of the first to emphasize the mental health consequences of involuntary full-time work. Avoiding role and time conflicts between family and work roles are important for the mental health of men too.

**Keywords:** work hour preferences; stress theory; family roles; household work; German socio-economic panel; conditional change models

## 1. Introduction

A growing group of scholars argues that we need to consider the match between the actual and preferred working hours of workers when studying the relation between working hours and health [1,2]. In particular, a mismatch can potentially negatively affect mental health, as it may signal a lack of control over the number of hours worked [3]. The reasons for wanting more or less hours may differ among workers (for instance, due to low income or high family responsibilities, etc.). Yet, these experiences (such as a sense of low income or high family responsibility) can also be present without the workers reporting a working hours mismatch [4]. In other words, workers with a working hours mismatch are a specific population that is characterized by a perceived lack of control over their working hours; i.e., a lack of schedule control [3]. In line with social stress theory [5], such a situation can lead to chronic or repeated stress reactions with negative consequences for mental health. Likewise, drawing on the Conservation of Resources (COR) theory [6], it can be argued that perceived schedule control represents a condition resource, that is, a condition that is valued by the

employee because it enables a favorable work-home interference and, therefore, wants to be preserved by the employee. Because of significant changes in the nature of the labor force (feminization of the labor force, the growing importance of leisure time, changing gender roles, etc.) the impact of the work-home balance on employees (and their mental health) is increasing [7]. Along these lines, the absence of resources protecting the work-home interference might exacerbate the negative effects of a sense of low schedule control on mental health.

In this study, we investigate the longitudinal relation of mental health and two involuntary working hour statuses, which we consider the most relevant in the contemporary labor market with workers seeking to find a satisfying balance between life roles: involuntary part-time employment, which is defined as working part-time but preferring to work full-time and involuntary full-time employment, which is defined as working full-time but preferring to work part-time. Additionally, because of the growing importance of a decent work-life balance, we also investigate the role of family- and work-related resources that might enhance a good work-life balance in the relationship between involuntary part- and full-time employment and mental health. These research questions will be investigated from a gender perspective; i.e., special attention is paid to the different gender roles of men and women both at home and in the workplace to justify our hypotheses.

*1.1. The Role of the Stress Process and Conservation of Resources for Workers' Mental Health*

Social stress theorists, such as Pearlin [8], Mirowsky and Ross [5], Aneshensel [9], and Thoits [10], put the insights of stress research in line with sociological interests. They argue that structural arrangements of people's lives (such as the work environment) affect a person's well-being through the "stress process" [8]. The stress process is the process where physical stress reactions arise from the strain of undesirable work conditions or arrangements, with a negative impact on mental health [11]. According to Mirowsky and Ross [5], an undesirable work condition (or stressor) causes stress reactions because that condition shapes the belief of the individual about "the nature of society, human relations, themselves, and their relationship to others and to society" and "the associated level of distress depends on the nature of these beliefs". As mentioned above, involuntary part- and full-time workers are a specific population that is characterized by a perceived lack of control over their working hours; i.e., a lack of schedule control [3]. The belief of control over one's own life is one of the most important beliefs about self and society that affect individual's health [5].

Moreover, drawing on the COR theory [6], which defines resources as objects (a house, a car), conditions (a good marriage, job stability), personal characteristics (self-esteem), or energy (money, favors), we argue that perceived schedule control can be defined as a condition resource. According to COR theory, stress occurs when an individual loses a resource, is threatened with a loss of resources, or fails to gain a resource following resource investment [6]. Moreover, COR theory emphasizes the gain and loss spiral. When an individual experiences stress they need to invest more resources because the stressor needs to be addressed. This is because individuals always attempt to obtain, retain, and protect resources [6]. This process of investment will continue, even if the coping is unsuccessful, until resources are depleted with severe stress reactions as a result (the loss spiral) [7].

A worker with a loss of a condition resource (schedule control) is at risk of losing other resources, such as a sense of mastery over one's (work) life and a favorable work-home interference. The worker will (invest to) protect and obtain these resources. On the one hand, being able to spend high-quality time on family, leisure, and socializing without the work-home balance being threatened gives rise to personal resources, such as a sense of mastery and self-esteem. On the other hand, a worker might invest time and energy in finding a job that has working hours in line with his/her preferences. When being unsuccessful in gaining fulfilling time outside of work or finding a job with a preferable number of working hours after investing time and energy, resources may be depleted and this can lead to chronic stress.

On the contrary, individuals with more resources have more opportunities to gain even more resources and are better equipped to deal with stressors; i.e., the gain spiral [7]. For instance, when

having a sense of high schedule control at work, workers do not need to invest in other resources to gain a sense of mastery, and a favorable work-home interference is (more) easily achieved.

In what follows, we argue that the stress caused by involuntary part-time and full-time work is partly the consequence of losing a resource; i.e., a sense of a lack of schedule control and partly the result of unsuccessfully investing resources to retain a sense of schedule control. This argument will be supported by considering the specific gender roles possessed by individuals.

### 1.1.1. Involuntary Part-Time Work

Employment is becoming increasingly non-standard, with an increasing number of part-time jobs [12]. Working part-time has the potential to increase time for leisure, socializing, and family. Yet, part-time jobs may insufficiently satisfy professional needs and aspirations [13]. Employers grant full-time workers, when compared to part-time workers, more frequently with high-quality job conditions (stable contract, socially protected, family wage, social transfers) that are regulated to a minimum level by collective agreement [14]. This is because part-time employment is commonly used as a flexibilization strategy by employers as a response to increased competition [15]. Thus, part-time jobs are more often jobs of lower quality and with lower financial rewards, compared to full-time jobs.

The quality and income of jobs are important determinants of worker mental health [16]. Nevertheless, previous research is inconsistent on whether full-time or part-time employment is better for mental health. Some research shows that working part-time is not associated with reduced mental health [17], while other studies report better mental health compared with full-time workers [12].

Inconsistent results might be the result of not considering whether workers want to work part- or full-time hours [4]. Yet, this group is growing as more and more workers are unable to find full-time opportunities, because full-time employment is more expensive for employers and part-time employment is used as a cheap labor [15]. Therefore, in this study, we distinguish regular part-time workers from part-time workers who want to work full-time (i.e., involuntary part-time workers). Low income or low-quality job characteristics are both present in regular part-time and involuntary part-time jobs, but the latter are a separate group characterized by a lack of schedule control that might affect their mental health.

Some studies exist on the mental health consequences of involuntary part-time work. A cross-sectional study found poor mental health for women working part-time who would prefer full-time employment [18]. The few previous longitudinal studies come to different conclusions. Dooley et al. [19] found that the involuntary part-time employed report significantly more depression than those adequately employed. A study based on the British Household Panel Survey found that, for men, wanting more hours when working part-time has no larger negative effect on psychological well-being than regular part-time employment [20]. This study also showed no worse psychological well-being for involuntary part-time employed women than for those full-time employed [20].

One explanation for inconsistent findings might be the different possibilities of men and women when working involuntary part-time (and thus when having low schedule control at work) to obtain schedule control in other domains of their life. Women usually have to work a double shift after work, taking care of children and the household [21], rendering them less time for protecting their sense of schedule control (e.g., looking for a job with their preferred number of working hours). On the contrary, men usually have a smaller role as care taker, and they are expected to be the breadwinner [21]. Therefore, when confronted with an hours mismatch, men will be able to obtain schedule control outside of work, either looking for another job or doing more fulfilling household work. We hypothesize that, controlled for type of job and financial background, involuntary part-time employment has a negative influence on female workers' mental health, while for men we expect no negative association (Hypothesis 1). We expect that it is not the lack of monetary and non-monetary gains that results in poor mental health, but their sense of low schedule control.

1.1.2. Involuntary Full-Time Work

Full-time work is the standard. Yet, the European work force went through profound changes due to, amongst other things, the feminization of the labor market after the 1960s. The accompanying demographic changes, namely the rise in dual-earner couples and single parent households [22], resulted in the blurring of traditional gender roles. In that regard, Aumann et al. [23] observed the growing preference of men to be more involved fathers, husbands/partners, and sons. But more generally too, shifts in employees' values and norms related to work time and time outside of work can be noticed [24,25]. Achieving a healthy or favorable balance between time at work, unpaid (care) work, and leisure time is increasingly important for employees [24]. This suggests that a preference for reduced hours can be seen among full-timers.

In the work environment, part-timers are seen as less dedicated and thus less professional [26]. As a consequence, in order not to damage future career prospects, a considerable number of workers work full-time but might actually prefer part-time employment. Moreover, financial reasons might also be an explanation why workers involuntarily stay in full-time employment. Men, especially, who are expected to be the breadwinner at home, might find themselves in this situation [21]. This phenomenon is referred to as involuntary full-time work [4].

Very little longitudinal research has taken involuntary full-time employment and its relation with mental health into account. There are two studies using a very broad sample of the working age population (including employed, unemployed, self-employed, and those out of the labor market (see Angrave and Charlwood [27] and Otterbach et al. [28]). Using UK data, Angrave and Charlwood [27] found that all types of overemployment when working full-time hours (i.e., working between 35-40h, 41–49h, and <49 h/week) results in lower psychological well-being, compared to those who prefer to work between 35 and 40 h [27]. These results are confirmed by Otterbach et al. [28] using the German Socio-Economic Panel.

Workers wanting to work less hours are those workers wanting to spend more time at home. Their employment situation prevents them from taking up their "care giver" roles, which seem to be increasingly important for men and women. Therefore, we hypothesize that involuntary full-time employment has a negative influence on workers' mental health (Hypothesis 2). Yet, as we expect that schedule control is the pathway between involuntary full-time work and mental health, the association is controlled for financial background.

*1.2. The Role of Family and Work-Related Resources*

As mentioned above, the loss spiral reflects the process in which stress develops and resources further deplete, while the gain spiral reflects a process in which resources accumulate [6]. Workers' mental health, when working involuntary part- and full-time hours, is thus expected to be protected when having resources that facilitate a sense of mastery and a favorable work-home interference. The presence of these resources might result in a gain spiral, while the absence of these resources might result in a loss spiral. In this study, we focus on those family and work-related resources that have the potential to protect a sense of mastery and a favorable work-home interference: social support, working standard hours, and doing a low number of household work hours.

Social support from the private life is an important resource that has proven to buffer work stress [29]. Mirowsky and Ross [5] define social support as the sense of being cared for and loved, being esteemed and valued as a person, and the sense of being part of a network of communication and obligation. According to COR theory, social support has two pathways through which they can facilitate a gain spiral. Firstly, social support from private life is a major potential route to resources that are beyond those possessed directly by individuals [6]. Close ties with others can provide meaningful aid in achieving, amongst other things, a sense of mastery and a new professional position, and can also provide help at the household level (financially or in the form of time spend on household labor). In other words, social support could result in the access to objects, conditions, personal characteristics, or energy resources that can help to protect or obtain a person's sense of mastery or

favorable work-home interference. Secondly, social support refers also to the identities of people. Individuals define themselves as husband, parent, etc. According to COR theory, people will also protect their identities [6]. The participation in multiple roles (such as parent and/or partner) has a beneficial effect on mental health as it can buffer distress in the worker role [22]. Because of the access to multiple identities and to resources beyond those possessed by the worker, we expect that the positive association of involuntary part-time work (for women) and full-time work (for men and women) with a deterioration of mental health will be less pronounced when having multiple family roles (Hypothesis 3).

Yet, time committed to household work takes time away from health-promoting activities. Time outside of (paid and unpaid) work is increasingly recognized as a resource because it enables healthy activities such as exercising, visiting a doctor, building strong and supportive relationships, etc. [30]. Household work is less rewarding than employment, it offers less recognition from others and offers a lower level of work fulfilment and a lower sense of control [31]. Doing household labor might even give rise to a sense of time pressure as there is less time for resource-promoting activities such as spending high-quality time with a significant other, looking for a better job, and leisure activities, resulting in a low sense of mastery and a dysfunctional work-home interference, leading to a loss spiral. So, the positive association of involuntary part-time work (for women) and full-time work (for men and women) with a deterioration of mental health will be less pronounced when spending less time on household labor (Hypothesis 4).

Moreover, family and social activities are traditionally organized outside of the standard work hours (during evenings and weekends). Therefore, working during non-standard hours can facilitate social marginalization and family dysfunction [32]. As explained previously, spending time with a significant other is a potential pool of personal resources (such as sense of belonging, sense of meaning, self-esteem, etc.). So, the positive association of involuntary part-time work (for women) and full-time work (for men and women) with a deterioration of mental health will be less pronounced when having standard hours (Hypothesis 5).

No study hitherto has examined the influence of family- and work-related resources on the effects of involuntary working hours. Only one explorative study exists, but it only included (in)voluntary full-time workers [4]. (In)voluntary part-time workers were not included in the study and, even though female workers were included, the results for female (in)voluntary full-time workers were not presented. This study suggests that, for involuntary full-time working men, additional social roles as a partner/husband and the amount of household workload influences mental health after two years [4]. Yet, no tests of significance were made to show whether these results were significantly different from their counterparts working regular full-time hours. Our study is innovative as it focusses on both (in)voluntary part- and full-time workers, and it formally tests the interaction between working hour categories and different types of family- and work-related resources. Primarily, combining COR theory and a gender perspective with the concept of working hours mismatch advances the literature, as these approaches are rarely used to explain the relation of working hours mismatch with health [33].

## 2. Materials and Methods

### 2.1. Data

Data were obtained from the German Socio-Economic Panel (GSOEP). This panel survey has been repeated yearly among households in Germany since 1984 and has collected information from more than 30,000 individuals aged 16 or above [34]. It provides data on a broad range of demographic, labor market, and health-related issues. For our analyses, we used data from waves 21 (2004), 23 (2006), 25 (2008), and 27 (2010) to create baseline observations for the measurement of involuntary part- or full-time work, family-related resources, and mental health (see Table 1). The outcome measure was based on information on mental health assessed two years later. Information on non-standard worktime was only available one year later compared to the baseline measurements (in waves 22, 24, 26, and 28,

respectively). This is not problematic because we excluded individuals who changed their job during one of the two-year-follow-up periods. We restricted the samples to individuals in paid work, aged 20–60 years. We excluded those in (partial) retirement, in-service training, military service, or voluntary service and those who were self-employed because they all have different employment relationships

**Table 1.** Description of baseline and follow-up data (German Socio-Economic Panel (GSOEP)).

| Baseline Assessment | Follow-Up [c] | N | | % | |
|---|---|---|---|---|---|
| | | Men | Women | Men | Women |
| 2004 [a] and 2005 [b] | 2006 | 3708 | 3674 | 29.0 | 28.8 |
| 2006 [a] and 2007 [b] | 2008 | 3506 | 3454 | 27.4 | 27.0 |
| 2008 [a] and 2009 [b] | 2010 | 3096 | 3076 | 24.2 | 24.1 |
| 2010 [a] and 2011 [b] | 2012 | 2485 | 2565 | 19.4 | 20.1 |
| Total (observations) | | 12,795 | 12,769 | 100 | 100 |
| Total (respondents) | | 5461 | 5622 | | |

[a] Involuntary part- or full-time work, family-related resources, mental health; [b] Work-related resources; [c] Mental health.

## 2.2. Measures

Mental health was assessed using the Mental Component Summary (MCS) score, a subscale from the Short Form 12 Health Survey, Version 2. The MCS has four subscales: (i) vitality, (ii) role limitation due to emotional problems, (iii) social functioning, and (iv) general mental health [35]. Two of the four subscales consist of one item each (i, iii); the other two consist of two items each (ii, iv). Results were represented by a value ranging from 0 to 100, with 0 representing the lowest level of mental health (and 100 the highest level). The scale was normalized using the German population in 2004 as a reference, with a mean of 50 and a standard deviation of 10 [35]. For the dependent variable, we subtracted the baseline value of the MCS score from the follow-up score two years later. This variable represents the difference in MCS score two years later.

We created a four-category working hour variable: (1) regular part-time: working and preferring < 35 h per week; (2) involuntary part-time: working < 35 h per week but preferring more hours; (3) regular full-time: working and preferring ≥ 35 h per week; and (4) involuntary full-time: working ≥ 35 h per week but preferring less hours. This variable was based on the number of working hours per week (overtime included) and the number of preferred hours per week, bearing in mind that earnings would go up or down according to the number of working hours chosen.

Family roles is a combination of the presence of children under the age of 14 in the household and the presence of a steady partner. This variable distinguishes between (living alone, single parent, partner with no children, and partner with children). Household workload is a combination of the hours spend on housework, errands, and childcare on a workday. We created a dummy variable separating 1 (those who spend more than the mean of their gender group on household work per workday) and 0 (those who spend less or equal to the mean of their gender group) (mean women = 5 h/day; mean men = 2 h per day). Standard worktime is a combination of four items on evening, night, Saturday, and Sunday work, each with five answer categories: No, Less frequently, Daily, Several times a week, and Alternating weeks. The answers were summarized (range 4 to 20) and divided into four groups: [1] Standard hours (range 4 to 5); [2] Somewhat non-standard hours (range 6 to 10); [3] High non-standard hours (range 11 to 15), and [4] Very high non-standard hours (range 16 to 20).

We additionally include three sociodemographic indicators (household income, type of job, and age groups), physical health, and survey year. A high income is related to better mental health [36] and to a lower willingness to work more hours [37]. Household income was rescaled in a three-level variable (low, medium, high) using tertiles. Household income is based on the monthly household income, which was adjusted for household size in accordance with the Organization for Economic Co-operation and Development (OECD) equivalent-scale [38]. People in high class jobs less frequently

report involuntary work hours [3] and depressive symptoms [39]. Type of job was grouped into four categories (based on the Erikson-Goldthorpe-Portocarero (EGP) scheme): "upper service class" (EGP I), "lower service class" (EGP II), "routine non-manual workers" (EGP III, IVab), and "skilled and unskilled manual workers" (EGP IVc, V, VI, VII). Age was grouped into four categories: "job starters" (20–29), "early midlife" (30–39), "late midlife" (40–49), and "older working life" (50–60). Job starters have better mental health than older workers [40]. Moreover, older workers are less likely to want more hours [41]. Because there is a strong link between mental and physical health [42], physical health was included in the models using the question "Does your health limit you in doing demanding everyday activities such as heavy lifting. Answer categories were "Greatly", "Somewhat", and "Not at all". Year distinguished the included baseline-waves of the GSOEP (2004, 2006, 2008, and 2010).

*2.3. Statistical Analysis*

All analyses were done for men and women separately. First, descriptive analyses were performed to present percentages, means, and standard deviations of all variables. Afterwards, conditional change models [43] were estimated using multilevel models for panel data. On the one hand, these models considered the hierarchical structure of the data; that is, that some observations (level 1) are not independent as they come from the same respondent nested in different survey years (level 2). On the other hand, conditional change models allow us to examine whether change over time in mental health is related to working hours at baseline adjusted for baseline mental health. A first model was estimated including the variables on working hours, family roles, baseline MCS scores, and all additional variables. Then Model 1 was extended by the interaction of the variable on working hours and family roles (Model 2), household workload (Model 3), and (non-)standard schedule (Model 4). By comparing models without and with interactions on the basis of a likelihood-ratio (LR) test, we tested for significant interactions. At all steps, parameter effects of the covariates in relation with change in mental health were presented as unstandardized regression coefficients (B). We applied complete case analyses, reducing the final sample to 10,266 observations for men (from 4596 men) and 9552 observations for women (from 4514 women). All calculations were done using STATA v14.2.

**3. Results**

Descriptive statistics are presented in Table 2. Among men, 7% report working involuntary full-time, compared to 14% among women. Only 2% of men report working involuntary part-time, compared to 5% among women. On average, women work 12 h per week less than men (44 h compared to 32 h). Men more frequently had (very) high non-standard hours, compared to women. For both men and women, less than 30% do more household work than their gendered average. Both men and women most frequently live with a partner (without children). Women more frequently report to be single parents, while men more frequently report to live with a partner and children.

**Table 2.** Description (in %) of the population studied (Population in salaried employment, 20–60 years old, 10,266 observations for men and 9552 observations for women, GSOEP wave 21–27).

|  | Men | | | Women | | |
|---|---|---|---|---|---|---|
|  | **N** | **%** | **Mean** | **N** | **%** | **Mean** |
| **Baseline MCS** | 10,266 | | 51.3 | 9552 | | 49.5 |
| **Follow-up MCS** | 10,266 | | 51.1 | 9552 | | 49.4 |
| **Actual working hours** | 10,266 | | 43.5 | 9552 | | 32.4 |
| **Desired working hours** | 10,266 | | 39.1 | 9552 | | 30.1 |
| **Work hours** | | | | | | |
| Regular part-time | 259 | 2.5 | | 3908 | 40.9 | |
| Involuntary part-time | 160 | 1.6 | | 499 | 5.2 | |
| Regular full-time | 9017 | 88.9 | | 3795 | 39.7 | |
| Involuntary full-time | 714 | 7.0 | | 1350 | 14.1 | |

**Table 2.** *Cont.*

| | Men | | | Women | | |
|---|---|---|---|---|---|---|
| | **N** | **%** | **Mean** | **N** | **%** | **Mean** |
| **Non-standard worktime** | | | | | | |
| No | 3454 | 33.7 | | 4789 | 50.1 | |
| Somewhat | 4137 | 40.3 | | 2999 | 31.4 | |
| High | 1719 | 16.7 | | 1236 | 12.9 | |
| Very high | 956 | 9.3 | | 528 | 5.5 | |
| **Family roles** | | | | | | |
| Alone | 1363 | 13.3 | | 1102 | 11.5 | |
| Single parent | 65 | 0.6 | | 255 | 2.7 | |
| Partner, no children | 5031 | 49.0 | | 5433 | 56.9 | |
| Partner, with children | 3807 | 37.1 | | 2762 | 28.9 | |
| **> gender mean on household work** | 2791 | 27.2 | | 2483 | 26.0 | |
| **Occupational class** | | | | | | |
| Higher service class | 4569 | 44.5 | | 3753 | 39.3 | |
| Lower service class | 1122 | 10.9 | | 3712 | 38.9 | |
| Routine non-manual work | 2654 | 25.9 | | 470 | 4.9 | |
| (Un)skilled manual work | 1921 | 18.7 | | 1617 | 16.9 | |
| **Physical health limitations** | | | | | | |
| Greatly | 455 | 4.4 | | 590 | 6.2 | |
| Somewhat | 3071 | 29.9 | | 3334 | 34.9 | |
| Not at all | 6740 | 65.7 | | 5628 | 58.9 | |
| **Age** | 10,266 | | 43.3 | 9552 | | 43.3 |
| **Monthly household income (euro)** | 10,266 | | 1900.1 | 9552 | | 1913.5 |
| **Baseline year** | | | | | | |
| 2004 | 2981 | 29.0 | | 2768 | 29.0 | |
| 2006 | 2761 | 26.9 | | 2533 | 26.5 | |
| 2008 | 2510 | 24.5 | | 2300 | 24.1 | |
| 2010 | 2014 | 19.6 | | 1951 | 20.4 | |

Abbreviation: MCS, Mental Component Summary

Table 3 presents the multilevel analyses for men. Model 1 shows that involuntary full-time work is significantly related to a negative change in mental health, compared with regular full-time work. Involuntary part-time work is not significantly related to a deterioration in mental health scores. In Model 2, we see that involuntary full-time work is significantly related to a negative change in mental health when living with a partner (with and without children), compared with regular full-timers who live alone. Part-time work for single fathers is associated with a positive change in mental health, but this finding is questionable due to the small number of single fathers in the sample. In Model 3, we see that involuntary part-time work is only significantly related to a negative change in mental health when having a high household workload. These workers report a reduction in MCS score of −1.73 points (= 0.68 − 2.48 + 0.07). This is larger than the reduction in MCS score reported by regular part-timers with high household labor: −1.57 points (= 0.06 − 1.70 + 0.07). In Model 4, we see that involuntary full-time work is significantly associated with a negative change in mental health when having high non-standard worktime: −3.40 points (= −1.97 − 0.88 − 0.55). This is a larger reduction than regular full-time work with high non-standard worktime: −0.55 points. The formal tests of the interactions show that there are significant differences in the relation between involuntary part-time/full-time work and deterioration of mental health depending on family roles and non-standard worktime and border on significant for household workload.

**Table 3.** Random effect models for difference in MCS scores in 10,266 observations for men (population in salaried employment, 20–60 years old, GSOEP wave 21–27. Models controlled for baseline MCS, year, age, income, occupational class, and physical health).

| | Model 1 | | Model 2 | | Model 3 | | Model 4 | |
|---|---|---|---|---|---|---|---|---|
| | **Beta** | **Sig.** | **Beta** | **Sig.** | **Beta** | **Sig.** | **Beta** | **Sig.** |
| Intercept | 22.9 | *** | 22.7 | *** | 22.8 | *** | 23.2 | *** |
| **Work hours (ref. = full-time)** | | | | | | | | |
| Part-time | −0.48 | | −1.08 | | 0.06 | | −1.09 | |
| Involuntary part-time | −0.23 | | −0.10 | | 0.68 | | −0.58 | |
| Involuntary full-time | −1.22 | *** | 1.12 | | −1.34 | *** | −0.88 | |
| **Family roles (ref. = Alone)** | | | | | | | | |
| Single parent | 0.92 | | 0.75 | | 0.91 | | 1.01 | |
| Couple, no kids | −0.23 | | −0.13 | | −0.21 | | −0.23 | |
| Couple, with children | −0.05 | | 0.09 | | −0.04 | | −0.04 | |
| **High household workload (ref. = Low)** | | | | | 0.07 | | | |
| **Non-standard worktime (ref. = No)** | | | | | | | | |
| Somewhat | | | | | | | −0.40 | * |
| High | | | | | | | −0.55 | * |
| Very high | | | | | | | −0.48 | |
| **Interactions** | | | | | | | | |
| **Work hours and family roles** | | | | | | | | |
| PT*single parent | | | 9.04 | * | | | | |
| PT*Couple, no kids | | | 0.68 | | | | | |
| PT*Couple, with kids | | | 0.48 | | | | | |
| Involuntary PT*single parent | | | −4.57 | | | | | |
| Involuntary PT*Couple, no kids | | | 0.68 | | | | | |
| Involuntary PT*Couple, with kids | | | −0.69 | | | | | |
| Involuntary FT*single parent | | | −5.51 | | | | | |
| Involuntary FT*Couple, no kids | | | −2.46 | * | | | | |
| Involuntary FT*Couple, with kids | | | −2.82 | ** | | | | |
| **Work hours and household workload** | | | | | | | | |
| PT*high | | | | | −1.70 | | | |
| Involuntary PT*high | | | | | −2.48 | * | | |
| Involuntary FT*high | | | | | 0.48 | | | |
| **Work hours and non-standard worktime** | | | | | | | | |
| PT*somewhat | | | | | | | 0.97 | |
| PT*High | | | | | | | 0.64 | |
| PT* Very high | | | | | | | 2.26 | |
| Involuntary PT*somewhat | | | | | | | 1.70 | |
| Involuntary PT*High | | | | | | | −2.89 | |
| Involuntary PT* Very high | | | | | | | 3.83 | |
| Involuntary FT*somewhat | | | | | | | −0.01 | |
| Involuntary FT*High | | | | | | | −1.97 | * |
| Involuntary FT* Very high | | | | | | | −0.80 | |
| Likelihood Ratio-test | | | 0.05 | | 0.06 | | 0.05 | |

* $p \leq 0.05$; ** $p \leq 0.01$; *** $p \leq 0.001$; Abbreviations; PT: Part-time; FT: Full-time; Sig.: Significance.

Table 4 presents the multilevel analyses for women. Model 1 show that regular part-time work is significantly related to a positive change in mental health compared with regular full-time work. There are no significant relations of involuntary part- or full-time work with deterioration of mental health. The interaction models show that there are no significant differences in the relation between involuntary part-time/full-time work and deterioration of mental health depending on the included family- and work-related resources (p-value of the LR-test comparing the model with and without interaction > 0.05). However, in Model 4 we see that involuntary full-time work is significantly related to a positive change in mental health when having somewhat non-standard worktime.

**Table 4.** Random effect models for difference in MCS scores in 9552 observations for women (population in salaried employment, 20–60 years old, GSOEP wave 21–27. Models controlled for baseline MCS, year, age, income, occupational class, and physical health).

| | Model 1 | | Model 2 | | Model 3 | | Model 4 | |
|---|---|---|---|---|---|---|---|---|
| | **Beta** | **Sig.** | **Beta** | **Sig.** | **Beta** | **Sig.** | **Beta** | **Sig.** |
| Intercept | 22.6 | *** | 22.6 | *** | 22.6 | *** | 22.9 | *** |
| **Work hours (ref. = full-time)** | | | | | | | | |
| Part-time | 0.52 | * | −0.16 | | 0.45 | | 0.13 | |
| Involuntary part-time | 0.69 | | 1.84 | | 0.71 | | 0.25 | |
| Involuntary full-time | −0.19 | | 0.04 | | −0.22 | | −0.88 | * |
| **Family roles (ref. = Alone)** | | | | | | | | |
| Single parent | −0.82 | | −1.09 | | −0.77 | | −0.86 | |
| Couple, no kids | −0.59 | * | −0.64 | | −0.58 | * | −0.60 | * |
| Couple, with children | −0.71 | * | −0.44 | | −0.69 | | −0.69 | * |
| **High household workload (ref. = Low)** | | | | | −0.32 | | | |
| **Non-standard work hours (ref. = No)** | | | | | | | | |
| Somewhat | | | | | | | −0.67 | * |
| High | | | | | | | −0.47 | |
| Very high | | | | | | | −1.08 | |
| **Interactions** | | | | | | | | |
| **Work hours and family roles** | | | | | | | | |
| PT*single parent | | | 0.97 | | | | | |
| PT*Couple, no kids | | | 0.89 | | | | | |
| PT*Couple, with kids | | | 0.33 | | | | | |
| Involuntary PT*single parent | | | 1.29 | | | | | |
| Involuntary PT*Couple, no kids | | | −1.11 | | | | | |
| Involuntary PT*Couple, with kids | | | −2.30 | | | | | |
| Involuntary FT*single parent | | | −1.07 | | | | | |
| Involuntary FT*Couple, no kids | | | −0.32 | | | | | |
| Involuntary FT*Couple, with kids | | | −0.03 | | | | | |
| **Work hours and household workload** | | | | | | | | |
| PT*high | | | | | 0.39 | | | |
| Involuntary PT*high | | | | | 0.16 | | | |
| Involuntary FT*high | | | | | 0.31 | | | |
| **Work hours and non-standard worktime** | | | | | | | | |
| PT*somewhat | | | | | | | 0.76 | |
| PT*High | | | | | | | 0.77 | |
| PT* Very high | | | | | | | 0.26 | |
| Involuntary PT*somewhat | | | | | | | 1.03 | |
| Involuntary PT*High | | | | | | | 0.12 | |
| Involuntary PT*Very high | | | | | | | 1.02 | |
| Involuntary FT*somewhat | | | | | | | 1.49 | * |
| Involuntary FT*High | | | | | | | 0.89 | |
| Involuntary FT*Very high | | | | | | | 1.09 | |
| Likelihood Ratio-test | | | 0.44 | | 0.91 | | 0.49 | |

* $p \leq 0.05$; *** $p \leq 0.001$; Abbreviations; PT: Part-time; FT: Full-time; Sig.: Significance.

## 4. Discussion

The aim of this study was to investigate whether involuntary part-time and involuntary full-time work is associated with a deterioration of mental health (assessed two years later) and whether this relation is buffered by family- and work-related resources (measured as family roles, low household workload, and standard worktime). This was investigated in a representative sample of German employees using data from the GSOEP.

The first important finding was that Hypothesis 1 can be rejected: For women, involuntary part-time work is not related to a reduction in mental health. Moreover, for women, no family- or work-related resource plays a significant role in this association. The former finding is in line with the results of Robone et al. [20], but it contradicts those of De Moortel et al. [18] and Dooley et al. [19]. Our result could be due to a large amount of heterogeneity among women working involuntary

part-time. For instance, some of them could be happy because they are complying with gender norm prescriptions, while others are unhappy because they are unable to meet professional needs [44].

In contrast to what was expected, we found no association between involuntary full-time work and a deterioration of mental health for women. Hypothesis 2 can thus be rejected for women. Our result contradicts the findings of Angrave and Charlwood [27] and Otterbach et al. [28]. This could be related to different samples used to study involuntary working hours. The above-mentioned studies investigated the working age population (including self-employed, unemployed, and those out of the labor market). Because, in our study, we focused on the resources increasing a good work-life balance, we restricted our sample to wage earners, a population group with limited power to demand a good work-life balance as they entered a specific employment relationship with an employer.

However, our findings could also be related to data limitations. To discover strong mental health effects of involuntary full-time work for women, the delay between the two measurement points might be too long. Because work-life balance is predominantly assumed to be an issue of working women, when confronted with working hours that are not in line with preferences, women might more quickly resolve it, compared to men [45].

Yet, involuntary full-time work is related to a deterioration of mental health for men (confirming Hypothesis 2). In line with previous research [27,28], not only are the actual working hours a determinant of mental health, but also preferences matter. In line with stress process theory, a low sense of schedule control might be the pathway whereby involuntary full-time work damages the mental health of male workers.

In contrast to what was expected (in Hypothesis 3), we found that the presence of a steady partner with or without children strengthens the association between involuntary full-time work and a deterioration of mental health for men. This result underscores the similar finding of De Moortel [4] by formally testing its significance. In all likelihood, involvement as a father, husband, or partner is important for men's mental health. Yet, social norms prescribe men to be the main provider when having a family [46]. Similarly to women, the combination of different social roles (worker, partner, father) might result in role stress, rather than acting as a pool of resources [47]. Especially for men who want less hours than full-time hours, the expectations of high engagement at work and in the family puts them at risk of poor mental health. Men's vulnerability to role conflict when wanting fewer hours is also supported by the finding that non-standard worktime strengthens the detrimental mental health effect of involuntary full-time work. In line with previous research [48], non-standard work hours hinder time spend with family and thus exaggerate the stress associated with involuntary full-time work. Thus, Hypothesis 5 can be confirmed for men.

Finally, in contrast to De Moortel [4], we found that having a low household workload does not protect involuntary full-time workers from poor mental health. Being able to combine an involuntary full-time job with a high household workload might be a sign that the work-home balance is not disturbed. Moreover, our results suggest that men in involuntary part-time work with a low household workload have better mental health, compared to their counterparts with a high household workload. Especially for men who want to work more hours, a high number of household work hours might threaten their "breadwinner" identity, causing distress [44]. Doing a high household workload might threaten their sense of schedule control. Perhaps these workers experience not enough time to find employment in line with their working hour preferences. So, Hypothesis 4 is partly confirmed for men.

Primarily, our findings suggest that regular part-time work for women is related to an increase in mental health, compared to regular full-time work. In all likelihood, these women benefit from more time for leisure, socializing, and other healthy activities [30].

This study has several limitations. Because mental health status was only asked every two years (since 2002), we were unable to reduce the time lag. Moreover, although our sample only included workers that stayed in the same job for two years, this does not mean that the level or presence of involuntary part- and full-time work remained stable during this period. The exclusion of respondents who changed jobs during the observational period might cause selection bias because those with no

work hours mismatch might be more likely to stay in the sample. This could lead to an underestimation of the mental health effects of involuntary part- and full-time work. Moreover, among regular part-time workers, some might want to work even less and can be considered a specific form of working hours mismatch (i.e., overemployed part-timers). Similarly, among regular full-time workers, some might want to work more and can also be considered a specific form of working hours mismatch (i.e., underemployed full-timers). In this study, we only considered two working hours mismatch statuses because these two are the most relevant in the contemporary labor market, and a small number of working hour categories make for more interpretable interactions. Another limitation is the use of family roles to measure social support. Previous research suggests that the belief that emotional support is available is a much stronger influence on mental health than the presence of a social roles [10]. Moreover, respondents' estimates about the hours per week typically spent on sets of related activities, such as (household) work, is subject to recall bias, typically leading to overestimates [30]. Lastly, the data used for investigating our research questions were collected in 2004, 2006, and 2008. Therefore, our results might also reflect the specific contextual and socioeconomic situation of that time.

The limitations are balanced by strengths. The use of conditional change models is one strength of this study. Involuntary part- and full-time workers have lower mental health at baseline, compared to workers who's actual and preferred hours match. The conditional change model is seen as an attempt to remove baseline differences between different groups of workers [43]. Moreover, the use of the GSOEP is a clear strength. The GSOEP provides a large longitudinal dataset representative of the German population. In addition, combining COR theory and a gender perspective with the concept of working hours mismatch has been shown to be useful in explaining the relation between involuntary working hours, family and job-related resources, and worker mental health.

## 5. Conclusions

This study adds to previous research by including the involuntary nature of working hours to the debate on the mental health impact of part- and full-time work. It was shown that involuntary full-time work for men in particular causes poor mental health. This study suggests that, just like working women, men's mental health is not safe from role conflicts. In contrast, while working, women's mental health might be protected by traditional gender norms prescribing them to be a care giver, male breadwinner norms might actually be harmful as they lead to role and time conflicts at a time where engagement in the family is becoming more important. These findings support recent policy proposals that include men (not just mothers) when discussing the need for more choice about how to balance work and family life [49]. More research is needed to confirm our results in other countries.

**Author Contributions:** Conceptualization, D.D.M., N.D. and M.W.; Formal analysis, D.D.M.; Funding acquisition, D.D.M.; Methodology, D.D.M. and M.W.; Software, D.D.M. and M.W.; Writing—original draft, D.D.M.; Writing—review & editing, D.D.M., N.D. and M.W. All authors have read and agreed to the published version of the manuscript.

**Funding:** Deborah De Moortel is a FWO [PEGASUS]² Marie Skłodowska-Curie Fellow. Her research has received funding from the FWO and European Union's Horizon 2020 research and innovation programme under the Marie Skłodowska-Curie grant agreement No 665501.

**Conflicts of Interest:** The authors declare no conflict of interest.

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
