# Peer review of "Involuntary Full- and Part-Time Work: Employees’ Mental Health and the Role of Family- and Work-Related Resources"

_societies, doi:10.3390/soc10040081_

Round 1

Reviewer 1 Report

First of all, I’d like to give my congratulations to the author/s for the manuscript and for giving me the opportunity to read it. The study is very interesting, with important results. However, I have some concerns as outlined below.

  • Which are the hypotheses of the study?
  • Please, give more information about the specific theory in which the study is based, and also about the procedure in the study: for example, about the ethical aspects of the study.
  • Authors should think about the convenience to control by type of professions, sector, level of studies, characteristics of the job (job demands, job resources…).
  • Why mental health is analyzed two years after? Maybe changes in mental health are consequence of other conditions and not by the involuntary full-and part-time work.
  • Which is the real novelty of the study? Which are the theoretical and practical implications?
  • A more current review of the literature should be performed.

I hope that these previous comments are interesting for authors.

Author Response

We would firstly like to thank the reviewer for taking the time to carefully read our manuscript. We appreciate the complements, but also the comments given to our manuscript. Below we reply to each comment separately:

1) We now formulate 5 hypotheses. These hypotheses are based upon the Conservation of Resources theory (see introduction of manuscript).

2) The introduction has been rewritten in order to have a more thorough theoretical background. We base our hypotheses on the stress process theory and the Conservation of Resources theory (COR) and we do this from a gender perspective.

When we cite “stress theory” in the original manuscript, we actually need to cite “social stress theory” to be correct. We now more correctly refer to “social stress theory” instead of stress theory and elaborated more on the stress process. Social stress theorists, such as Pearlin (1989); Mirowsky & Ross (1986); Aneshensel (1992); Thoits, (1995), put the insights of stress research in line with sociological interests. They argue that structural arrangements of people’s lives (such as the work environment) affect a person’s well-being through the “stress process” [1]. The stress process is the process where stress arises from the strain of undesirable work conditions or arrangements, which may ultimately bear on mental health [5]. According to Mirowsky & Ross (1986), an undesirable work condition (or stressor) causes stress because that condition shapes the belief of the individual about “the nature of society, human relations, themselves, and their relationship to others and to society and the associated level of distress depends on the nature of these beliefs”. Workers with a working hours mismatch are a specific population that is characterized by a lack of control over their working hours, i.e. a lack of schedule control [6]. The belief in control over one’s own life is one of the most important beliefs about self and society that affect individual’s distress [2]. So, in our manuscript, it is argued that the strain caused by involuntary part-time and full-time work is a consequence of a sense of a lack of schedule control and that the associated stress results in poor mental health.

Moreover, drawing on the Conservation of Resources (COR) theory [7], which defines resources as objects (a house, a car), conditions (a good marriage, job stability), personal characteristics (self-esteem) or energy (money, favours), we argue the belief of schedule control can be defined as a condition resource, namely a condition that is valued by the employee to gain a favourable work-home interference. Because of dramatic changes in the nature of the labour force (feminization of the labour force, the growing importance of leisure time, changing gender roles, etc.) the impact of the work-home interference on employees (and their mental health) is increasing [8].

According to COR theory, stress occurs when an individual loses a resource, is threatened with a loss of resources, or fails to gain a resource following resource investment [7]. Moreover, COR theory emphasis on the gain and loss spiral. When an individual experience stress they need to invest more resources because the stressor needs to be addressed. This is because individuals always attempt to obtain, retain, and protect resources [7]. This process of investment will continue, even if the coping is unsuccessful, until resources are depleted with severe stress as a result (the loss spiral). For instance, a worker with a loss of a condition resource (schedule control) is threatened to lose other resources, such as a sense of mastery over one’s (work) life and a favourable work-home interference. The worker will (invest to) protect and obtain these resources. On the one hand, being able to spend high-quality time on family, leisure and socializing without the work-home interference being threatened, gives rise to the personal resources, such as sense of mastery, self-esteem. On the other hand, a worker might invest time and energy in finding a job that has working hours in line with his/her preferences. When being unsuccessful in gaining fulfilling time outside of work or finding a job with a preferable number of working hours after investing time and energy, resources might be depleted and severe stress will be the result.

On the contrary, individuals with more resources have more opportunities to invest in gaining more resources and are better equipped to deal with stressors, i.e. the gain spiral [8]. For instance, when having a sense of high schedule control at work, workers do not need to invest in other resources to gain a sense of mastery and a favourable work-home interference is (more) easily gained. In sum, the loss spiral reflects the process in which stress develops and resources further deplete, while the gain spiral reflects a process in which resources accumulate [7].

Workers’ mental health when working involuntary part- and full-time hours, is thus expected to be protected when having resources that facilitate a sense of mastery and a favourable work-home interference. The presence of these resource might result in a gain spiral, while the absence of these resource might result in a loss spiral. In this study, we focus on those family and work-related resources that have the potential to protect a sense of mastery and a favourable work-home interference: social support, working standard hours and doing a low number of household work hours.

The study was approved by the ethical committee of the Medical Department of the Heinrich Heine University of Düsseldorf. We performed a secondary data analysis on pre-existing data.

3) Our models are controlled household income, type of job, age groups, physical health, survey year. We did not control for education, because our models are already controlled for income and type of job. These two variables are already two indicators of social class. By including education, we introduce a third indicator of social class to our models, which is to our opinion unnecessary. The same reasoning can be applied to adding the variables type of professions or sector. Including the indicator type of profession is very similar to the already included control variable “type of job”. Moreover, we did not include additional control variables of job characteristics because of the control variable type of job was already included in the study. We did not include other job resources as this would be too confusing as the purpose of our study is to investigate the role of job resources for the relation between involuntary part-time and full-time work on the one hand and mental health on the other hand. When including more job resources, we believe we need to theoretically justify the inclusion of these job resources. This would only lengthen the already lengthy manuscript.

4) We only analyzed mental health after two years, and not after one year, because we are limited to our data. Mental health is only measured bi-annually. Thus, unfortunately we could not reduce the time-lag of our study. We point at this limitation of our study in the discussion section. Another reason to measure mental health after two year is to proof the causal relation between the working hours statuses and mental health.

The reasons for changes in mental health are now better explained in the introduction. We used the stress process theory and the conservation of resources theory to explain the relation between involuntary working hours and mental health.

4) One of the novelties of the studies is the use of the stress process theory, the Conservation of Resources theory from a gender perspective applied to the field of working hours mismatch. This is now explicitly mentioned in the revised manuscript.

5) We added some recent studies to the revised manuscript, some examples:

 Peckham, Fujishiro, Hajat, Flaherty, Seixas. Evaluating Employment Quality as a Determinant of Health in a Changing Labor Market. RSF Russell Sage Found J Soc Sci. 2019;5:258.

Vanroelen C. Employment Quality: An Overlooked Determinant of Workers’ Health and Well-being? Ann Work Expo Heal. 2019;63:619–23.

 Moreover, the studies included by the authors (referred to as Anonymous in the manuscript) are also very recent studies.

By completely rewriting the introduction, we hope the current literature is sufficiently included.

Reviewer 2 Report

I would like to thank the authors for their effort in developing this Manuscript:
“Involuntary full- and part-time work: employees’ mental health and the role of family- and work-3 related resources”. This manuscript presents an interesting idea and uses strong research design, however, based on my reading it needs to be developed further before it can be published. In the following section, I provide some comments and suggestions to the authors aiming they could be helpful to strengthen their contribution.

Introduction
In the introduction, the authors present some empirical background regarding the relationship between the two involuntary working statuses and mental health. However, a lack of theoretical elaboration is provided. The authors cite “stress theory” in several parts of the manuscript, but it is not clear which stress theory they are referring to (you have to look into the references to find that out). There are many stress theories, many of which explore the role of resources (for example Conservation of Resources Theory, Hobfoll, 1989; the Job Demands Resources Model (Demerouti et al., 2001). It seems crucial that the authors can explain their hypothesis based on one specific theory of stress. More importantly, the introduction needs additional details on how the proposed moderators are supported both empirically and theoretically. Significance of the study should be also made more explicit: how does this study advance the literature on this topic? This seems particularly important since the results show some of the same inconsistencies as previous studies in this area.

Besides, there is a lack of justification for gender differences in the proposed relations; however, the results and discussion sections are entirely based on exploring gender differences.
Some arguments used to justify the possible difference in mental health due to the discrepancy in the working hour status, maybe explain by other variables, processes, or mechanisms.

For example on page 2, the authors indicate: “This suggests that a preference for reduced hours can be seen among full-timers. Yet, in the work environment part-timers are seen as less dedicated and thus less professional [11]. As a consequence, in order not to damage future career prospects, a considerable number of workers work full-time, but might actually prefer part-time employment.”
In this case, financial reasons (stressors) might play an important role in explaining why people remain at full-time work.

Similarly, they argue the following (page 2): “Few studies exist on the mental health consequences of involuntary part-time and full-time work. A cross-sectional study found poor mental health for women working part-time, but preferring full-time employment”
However, I wonder if this lower mental health was due to the discrepancy in the working hour status or any other reason (lack of job quality, financial pressure, etc).

Therefore, the introduction lacks a solid conceptualization of why the discrepancy is related to mental health. Previous results are inconsistent, and many other cofounding variables and explanation mechanisms are not addressed.

I recommend the authors to consider present separately the argumentation for the two different groups studied here: involuntary part-time and involuntary full time. Based on their arguments, it seems that both types of situations have some advantages and disadvantages for mental health.
I would recommend elaborating on both in separate sections and to strengthen the theoretical background that supports how these two kinds of involuntarily work status affect mental health, and how and why would these differences not be equal between men and women. Besides, they should support their theoretical argumentation of the role of resources as buffers.

Methodology & Results
The authors used a large data set with a longitudinal design to test their hypotheses (which are not explicitly presented in the introduction). However, data analyzed was collected almost ten years ago (the nearest lag); thus contextual and socioeconomic situations might be different. I think this should be addressed in the limitation section.

Discussion
Results show some of the inconsistencies already presented in the available literature. Significant and non-significant results were not deeply explained based on any theoretical model of stress or other justification. Discussion of the results was mostly centered on gender differences, while no gender differences were included in the introduction. It seems that the effect of involuntary work status affects mental health differently by gender. So, gender seems to be s contingency of the effect of involuntary work status, as a stressor, on mental health. Nevertheless, the authors should provide a clear justification of why is this possible and to present a solid argument about this difference.

Author Response

We are grateful to reviewer 2 for taking the time to carefully read our manuscript. We like to thank the reviewer for highlighting that our manuscript presents an interesting idea and uses a strong research design. We are furthermore grateful for the comments received by the reviewer, as they helped us to further improve our manuscript. Below we reply to each comment separately:

1) The term resources, as noted by REVIEWER 2, pointed in the direction of Job Demand Resources (JD-R) theory or Conservation of Resources (COR) Theory. Yet, two of the three characteristics (household workload and non-standard work hours) are not part of the traditional resources of the JD-R model. In the model of Karasek et al. (1998), resources refer to intrinsic job characteristics, or characteristics related to the job task, such as autonomy, skill discretion, etc. Moreover, in the JD-R model resources related to the “family domain” are underrepresented (with the exception of social support). Yet, family-related resources are important when investigating a sense of low schedule control (the mechanism explaining poor health for workers with an hours mismatch), which is closely related to the work-nonwork interference (Schieman, Milkie, & Glavin, 2009). The COR theory looks both at resource from the work and family domain (Hobfoll & Shirom, 2000). Drawing on the Conversation of Resources (COR) theory, we call the family and job characteristics used in our manuscript “resources”. Combining the COR theory with the concept of working hours mismatch is one aspect of our manuscript which advances the literature of working hour mismatches, as this is rarely used to explain its relation with health (Maynard & Feldman, 2011).

When we cite “stress theory” in the original manuscript, we actually need to cite “social stress theory” to be correct. We now more correctly refer to “social stress theory” instead of stress theory and elaborated more on the stress process. Social stress theorists, such as Pearlin (1989); Mirowsky & Ross (1986); Aneshensel (1992); Thoits, (1995), put the insights of stress research in line with sociological interests. They argue that structural arrangements of people’s lives (such as the work environment) affect a person’s well-being through the “stress process” (Pearlin, 1989). The stress process is the process where stress arises from the strain of undesirable work conditions or arrangements, which may ultimately bear on mental health (Tompa, Scott-Marshall, Dolinschi, Trevithick, & Bhattacharyya, 2007). According to Mirowsky & Ross (1986), an undesirable work condition (or stressor) causes stress because that condition shapes the belief of the individual about “the nature of society, human relations, themselves, and their relationship to others and to society and the associated level of distress depends on the nature of these beliefs”. Workers with a working hours mismatch are a specific population that is characterized by a lack of control over their working hours, i.e. a lack of schedule control (Lyness, Gornick, Stone, & Grotto, 2012). The belief in control over one’s own life is one of the most important beliefs about self and society that affect individual’s distress (Mirowsky & Ross, 1986). So, in our manuscript, it is argued that the strain caused by involuntary part-time and full-time work is a consequence of a sense of a lack of schedule control and that the associated stress results in poor mental health.

Moreover, drawing on the Conservation of Resources (COR) theory (Hobfoll & Shirom, 2000), which defines resources as objects (a house, a car), conditions (a good marriage, job stability), personal characteristics (self-esteem) or energy (money, favours), we argue the belief of schedule control can be defined as a condition resource, namely a condition that is valued by the employee to gain a favourable work-home interference. Because of dramatic changes in the nature of the labour force (feminization of the labour force, the growing importance of leisure time, changing gender roles, etc.) the impact of the work-home interference on employees (and their mental health) is increasing (ten Brummelhuis & Bakker, 2012).

According to COR theory, stress occurs when an individual loses a resource, is threatened with a loss of resources, or fails to gain a resource following resource investment (Hobfoll & Shirom, 2000). Moreover, COR theory emphasis on the gain and loss spiral. When an individual experience stress they need to invest more resources because the stressor needs to be addressed. This is because individuals always attempt to obtain, retain, and protect resources (Hobfoll & Shirom, 2000). This process of investment will continue, even if the coping is unsuccessful, until resources are depleted with severe stress as a result (the loss spiral). For instance, a worker with a loss of a condition resource (schedule control) is threatened to lose other resources, such as a sense of mastery over one’s (work) life and a favourable work-home interference. The worker will (invest to) protect and obtain these resources. On the one hand, being able to spend high-quality time on family, leisure and socializing without the work-home interference being threatened, gives rise to the personal resources, such as sense of mastery, self-esteem. On the other hand, a worker might invest time and energy in finding a job that has working hours in line with his/her preferences. When being unsuccessful in gaining fulfilling time outside of work or finding a job with a preferable number of working hours after investing time and energy, resources might be depleted and severe stress will be the result.

On the contrary, individuals with more resources have more opportunities to invest in gaining more resources and are better equipped to deal with stressors, i.e. the gain spiral (ten Brummelhuis & Bakker, 2012). For instance, when having a sense of high schedule control at work, workers do not need to invest in other resources to gain a sense of mastery and a favourable work-home interference is (more) easily gained. In sum, the loss spiral reflects the process in which stress develops and resources further deplete, while the gain spiral reflects a process in which resources accumulate (Hobfoll & Shirom, 2000).

Workers’ mental health when working involuntary part- and full-time hours, is thus expected to be protected when having resources that facilitate a sense of mastery and a favourable work-home interference. The presence of these resource might result in a gain spiral, while the absence of these resource might result in a loss spiral. In this study, we focus on those family and work-related resources that have the potential to protect a sense of mastery and a favourable work-home interference: social support, working standard hours and doing a low number of household work hours.

Social support from the private life is an important resource which has proven to buffer work stress [29]. Mirowsky & Ross [5] define social support as the sense of being cared for and loved, being esteemed and valued as a person, and the sense of being part of a network of communication and obligation. According to COR theory, social support has two pathways through which they can facilitate a gain spiral. Firstly, social support from the private life is a major potential route to resources that are beyond those possessed directly by individuals [6]. Ties with close others can provide meaningful aid in achieving amongst others a sense of mastery, a new professional position, provide help at the household level (financially or in the form of time spend on household labour). In other words, social support could result in the access to objects, conditions, personal characteristics or energy resources able to protect or obtain a person’s sense of mastery or favourable work-home interference. Secondly, social support refers also to the identities of people. Individuals define themselves as husband, parent, etc. According to COR theory, people will also protect their identities [6]. The participation in multiple roles (such as parent and/or partner) has a beneficial effect on mental health as it can buffer distress in the worker role [22]. Because of the access to multiple identities and to resources beyond those possessed by the worker, we expect that the positive association between mental health on the one hand and involuntary part-time work (for women) and full-time work (for men and women) on the other hand will be less pronounced when having multiple family roles (Hypothesis 3).

Yet, time committed to household work takes time away from health-promoting activities. Time outside of (paid and unpaid) work is increasingly recognised as a resource, because it enables healthy activities such as exercising, visiting a doctor, building strong and supportive relationships, etc. [30]. Household work is less rewarding than employment, it offers less recognition from others, and offers a lower level of work fulfilment and lower sense of control [31]. Doing household labour might even give rise to a sense of time pressure as there is less time for resource-promoting activities, such as spending high-quality time with significant other, looking for a better job, leisure activities, resulting in low sense of mastery and a dysfunctional work-home interference, leading to a loss spiral. So, the positive association between mental health on the one hand and involuntary part-time work (for women) and full-time work (for men and women) on the other hand will be less pronounced when spending less time on household labour (Hypothesis 4).

Moreover, family and social activities are traditionally organised outside of the standard work hours (during evenings and weekends). Therefore, working during non-standard hours can facilitate social marginalisation and family dysfunction [32]. As explained spending time with significant other is a potential pool of personal resources (such as sense of belonging, sense of meaning, self-esteem, …). So, the positive association between mental health on the one hand and involuntary part-time work (for women) and full-time work (for men and women) on the other hand will be less pronounced when having standard hours (Hypothesis 5).

The introduction of the manuscript has been rewritten to specifically mention the COR theory, the hypotheses and the advances for the literature.

2) Thank you for pointing our attention to the lack of attention to gender relations in the introduction. The introduction of the manuscript has been rewritten to address gender issues in the relations under study.

See page 2: These research questions will be investigated using a gender perspective, i.e. special attention is paid to the different gender roles of men and women both at home and in the workplace to justify our hypotheses.

See page 2: This argument will be supported by considering the specific gender roles possessed by individuals.

See page 4: One explanation for inconsistent findings might be the different possibilities of men and women, when working involuntary part-time (and thus when having low schedule control at work), to obtain schedule control in other domains of their life. Women usually have to work a double shift after work, taking care of children and the household [22]. Rendering them less time for protecting their sense of schedule control (e.g. looking for a job with their preferred number of working hours). On the contrary, men usually have a smaller role as care taker and they are expected to be the breadwinner. Therefore when confronted with an hours mismatch, men will be able to obtain schedule control outside of work, either looking for another job or doing more fulfulling hoursehold work. We hypothesize that controlled for job characteristics and financial background, involuntary part-time employment has a negative influence on female workers’ mental health, while for men we expect no negative association (Hypothesis 1). 

See page 5: The accompanying demographic changes, namely the rise in dual-earner couples and single parent households [23], resulted in the blurring of traditional gender roles. In that regard, Aumann et al. [24] observed the growing preference of men to be a more involved father, husband/partner and son. But more generally too, shifts in employees’ values and norms related to work time and time outside of work can be noticed [25,26]. Achieving a healthy or favourable balance between time at work, unpaid (care) work and leisure time is increasingly important for employees [25].

3) We hope by elaborating on social stress theory and the Conservation of Resources theory that the unclarity about the mechanisms causing poor health when working involuntary part- or full-time work is addressed.

We argue that it is not lack of financial rewards or lack of job quality that causes poor mental health in involuntary part-time workers. These conditions are namely also present when working part-time voluntarily. The lack of job quality and low wages are a problem of part-time employment in general. Involuntary part-time workers are a specific group of the working population that experience a lack of schedule control. This is how regular part-time workers differentiate themselves from involuntary part-time workers. However, just to be safe we control our models for type of job and household income. This is now explicitly mentioned in the hypothesis.

We agree that involuntary full-time workers may stay in full-time employment because of financial reasons. A sentence has been added to highlight this in the manuscript. 

4) In the revised manuscript, we explain the stress process for involuntary part- and full-time workers separately as asked by REVIEWER 2. The theoretical background that support how these two involuntary work statuses affect health has been strengthened using the Conservation of Resources theory. The role of the resources is now supported by the Conservation of Resources theory and the loss and gain spiral (see introduction of the manuscript).

5) We now formulate 5 hypotheses. These hypotheses are based upon the Conservation of Resources theory (see introduction of manuscript).

We added a sentence to the limitations section of the discussion (see page 7): “Lastly, the data used for investigating our research questions were collected in 2014. Therefore, our results might also reflect the specific contextual and socioeconomic situation of that time.”

6) The introduction was rewritten to explain our results based on the COR theory with a gender perspective. By doing so we hope that the discussion of the results is now better supported by theory. Additionally, we added some words and sentences to bring our discussion in line with the arguments made in the introduction.

Reviewer 3 Report

The article deals with issues of interest, in fact, in the presented version rarely undertaken in literature. Methodological assumptions, adopted models, research issues have been properly prepared and formulated. However, sometimes the presented criteria of assigning certain subjects to the models seem to be not clear enough (perhaps it is a question of translation from German). One can get the impression that the authors did not use the obtained data or lacked editorial energy, which was visible in the initial parts of the article. According to the reviewer, the application values of the obtained research results should also be indicated, which would increase the value of the reviewed article.

Author Response

Firstly, we would like to thank REVIEWER 3 for the suggestions to improve the manuscript. We adjusted our manuscript in line with the comments made by the reviewer.

1) We agree with the reviewer that we lack a justification for the selection of the moderator variables (social support, nonstandard work time, household work load). Yet, we completely rewritten the introduction of the paper including the stress process theory, the Conservation of resources theory and a gender perspective.

This section has been added to explain the choice of moderators better:

“Drawing on the Conservation of Resources (COR) theory [6], which defines resources as objects (a house, a car), conditions (a good marriage, job stability), personal characteristics (self-esteem) or energy (money, favours), we argue that perceived schedule control can be defined as a condition resource. According to COR theory, stress occurs when an individual loses a resource, is threatened with a loss of resources, or fails to gain a resource following resource investment [6]. Moreover, COR theory emphasis on the gain and loss spiral. When an individual experience stress they need to invest more resources because the stressor needs to be addressed. This is because individuals always attempt to obtain, retain, and protect resources [6]. This process of investment will continue, even if the coping is unsuccessful, until resources are depleted with severe stress reactions as a result (the loss spiral) [7].

A worker with a loss of a condition resource (schedule control) is threatened to lose other resources, such as a sense of mastery over one’s (work) life and a favourable work-home interference. The worker will (invest to) protect and obtain these resources. On the one hand, being able to spend high-quality time on family, leisure and socializing without the work-home balance being threatened, gives rise to personal resources, such as sense of mastery and self-esteem. On the other hand, a worker might invest time and energy in finding a job that has working hours in line with his/her preferences. When being unsuccessful in gaining fulfilling time outside of work or finding a job with a preferable number of working hours after investing time and energy, resources may be depleted and lead to chronic stress.

On the contrary, individuals with more resources have more opportunities to gain even more resources and are better equipped to deal with stressors, i.e. the gain spiral [7]. For instance, when having a sense of high schedule control at work, workers do not need to invest in other resources to gain a sense of mastery and a favourable work-home interference is (more) easily acheived.

In what follows, we argue that the stress caused by involuntary part-time and full-time work is partly the consequence of losing a resource, i.e. a sense of a lack of schedule control and partly the result of unsuccesfully investing resources to retain a sense of schedule controle. This argument will be supported by considering the specific gender roles possessed by individuals.”

“The loss spiral reflects the process in which stress develops and resources further deplete, while the gain spiral reflects a process in which resources accumulate [6]. Workers’ mental health when working involuntary part- and full-time hours, is thus expected to be protected when having resources that facilitate a sense of mastery and a favourable work-home interference. The presence of these resources might result in a gain spiral, while the absence of these resources might result in a loss spiral. In this study, we focus on those family and work-related resources that have the potential to protect a sense of mastery and a favourable work-home interference: social support, working standard hours and doing a low number of household work hours.

Social support from the private life is an important resource which has proven to buffer work stress [29]. Mirowsky & Ross [5] define social support as the sense of being cared for and loved, being esteemed and valued as a person, and the sense of being part of a network of communication and obligation. According to COR theory, social support has two pathways through which they can facilitate a gain spiral. Firstly, social support from the private life is a major potential route to resources that are beyond those possessed directly by individuals [6]. Ties with close others can provide meaningful aid in achieving amongst others a sense of mastery, a new professional position, provide help at the household level (financially or in the form of time spend on household labour). In other words, social support could result in the access to objects, conditions, personal characteristics or energy resources able to protect or obtain a person’s sense of mastery or favourable work-home interference. Secondly, social support refers also to the identities of people. Individuals define themselves as husband, parent, etc. According to COR theory, people will also protect their identities [6]. The participation in multiple roles (such as parent and/or partner) has a beneficial effect on mental health as it can buffer distress in the worker role [22]. Because of the access to multiple identities and to resources beyond those possessed by the worker, we expect that the positive association between mental health on the one hand and involuntary part-time work (for women) and full-time work (for men and women) on the other hand will be less pronounced when having multiple family roles (Hypothesis 3).

Yet, time committed to household work takes time away from health-promoting activities. Time outside of (paid and unpaid) work is increasingly recognised as a resource, because it enables healthy activities such as exercising, visiting a doctor, building strong and supportive relationships, etc. [30]. Household work is less rewarding than employment, it offers less recognition from others, and offers a lower level of work fulfilment and lower sense of control [31]. Doing household labour might even give rise to a sense of time pressure as there is less time for resource-promoting activities, such as spending high-quality time with significant other, looking for a better job, leisure activities, resulting in low sense of mastery and a dysfunctional work-home interference, leading to a loss spiral. So, the positive association between mental health on the one hand and involuntary part-time work (for women) and full-time work (for men and women) on the other hand will be less pronounced when spending less time on household labour (Hypothesis 4).

Moreover, family and social activities are traditionally organised outside of the standard work hours (during evenings and weekends). Therefore, working during non-standard hours can facilitate social marginalisation and family dysfunction [32]. As explained spending time with significant other is a potential pool of personal resources (such as sense of belonging, sense of meaning, self-esteem, …). So, the positive association between mental health on the one hand and involuntary part-time work (for women) and full-time work (for men and women) on the other hand will be less pronounced when having standard hours (Hypothesis 5).”

2) One of the novelties of the studies is the use of the stress process theory, the Conservation of Resources theory from a gender perspective applied to the field of working hours mismatch. This is now explicitly mentioned several times in the revised manuscript.

Round 2

Reviewer 1 Report

The article has been improved according to the previous comments.

Author Response

Dear Reviewer 1, 

I would like to thank you for taking the time to read and approve the resubmitted manuscript.

There were no additional comments, so a point by point answer to comments is not necessary.

Kind regards, 

The authors.

Reviewer 2 Report

Dear authors,

I appreciate the effort and changes completed in the manuscript, especially to clarify and strengthen the theoretical background used to develop the hypothesis.

There is a last comment/suggestion. When stating Hypothesis 3 to 5, I would suggest inverting the order of the variables in the sentence to make clearer that the dependent variable in the model is mental health.

For example, in Hypothesis 3:

…the positive association between involuntary part-time work (for women) and full-time work (for men and women) on mental health will be less pronounced when having multiple family roles (Hypothesis 3).

All the additional suggestions were addressed. Thanks to the authors for this contribution.

Author Response

Dear Reviewer 2,

When stating Hypothesis 3 to 5, we inverted the order of the variables in the sentence to make clearer that the dependent variable in the model is mental health.

Hypothesis 3:

we expect that the positive association between involuntary part-time work (for women) and full-time work (for men and women) and deterioration of mental health will be less pronounced when having multiple family roles.   Hypothesis 4:    the positive association between involuntary part-time work (for women) and full-time work (for men and women) and deterioration of mental health will be less pronounced when spending less time on household labour.   Hypothesis 5:    the positive association between involuntary part-time work (for women) and full-time work (for men and women) and deterioration of mental health will be less pronounced when having standard hours.   We would like to thank the reviewer for taking the time to read our revised manuscript.   Kind regards,  The authors.